# An assembly-free method of phylogeny reconstruction using short-read sequences from pooled samples without barcodes

**Thomas K. F. Wong**[1]*, **Teng Li**[1,2], **Louis Ranjard**[1,3], **Steven H. Wu**[4], **Jeet Sukumaran**[5], **Allen G. Rodrigo**[1,2]*

**1** The Research School of Biology, The Australian National University, ACT, Australia, **2** School of Biological Sciences, University of Auckland, Auckland, New Zealand, **3** PlantTech Research Institute, Tauranga, New Zealand, **4** Department of Agronomy, National Taiwan University, Taipei, Taiwan, **5** Biology Department, San Diego State University, San Diego, California, United States of America

* thomas.wong@anu.edu.au (TKFW); a.rodrigo@auckland.ac.nz (AGR)

## Abstract

A current strategy for obtaining haplotype information from several individuals involves short-read sequencing of pooled amplicons, where fragments from each individual is identified by a unique DNA barcode. In this paper, we report a new method to recover the phylogeny of haplotypes from short-read sequences obtained using pooled amplicons from a mixture of individuals, without barcoding. The method, AFPhyloMix, accepts an alignment of the mixture of reads against a reference sequence, obtains the single-nucleotide-polymorphisms (SNP) patterns along the alignment, and constructs the phylogenetic tree according to the SNP patterns. AFPhyloMix adopts a Bayesian inference model to estimate the phylogeny of the haplotypes and their relative abundances, given that the number of haplotypes is known. In our simulations, AFPhyloMix achieved at least 80% accuracy at recovering the phylogenies and relative abundances of the constituent haplotypes, for mixtures with up to 15 haplotypes. AFPhyloMix also worked well on a real data set of kangaroo mitochondrial DNA sequences.

## Author summary

In evolutionary studies, it is customary to obtain homologous sequences from different individuals in a population or a species to construct a phylogeny. Frequently, sequences from different individuals will be identical; we refer to a set of identical sequences as a haplotype. If short-read sequencing technologies are used to obtain sequences from many individuals, the sequence from each individual is tagged with a unique barcode, and a mixed sample of tagged sequences is subsequently sequenced. The tagged sequences can be identified using the appropriate bioinformatics tools, for further downstream analyses. We have developed a novel method, AFPhyloMix, to reconstruct the phylogeny of a mixed sample of homologous sequences, and the relative abundance of different haplotypes, from different individuals without the need for barcoding. AFPhyloMix aligns the short reads obtained to a reference alignment, and identifies the variable sites along the

**Data Availability Statement:** The software AFPhyloMix and the materials are available in OSF

repository: https://osf.io/w5s2h/ (DOI: 10.17605/
OSF.IO/W5S2H).

**Funding:** This project was funded by an Australian
Research Council Discovery Project Grant
DP160103474 to AR and National Natural Science
Foundation of China No. 31501879 to TL. The
funders had no role in study design, data collection
and analysis, decision to publish, or preparation of
the manuscript.

**Competing interests:** The authors have declared
that no competing interests exist.

alignment. On the basis of the patterns of nucleotide frequencies at these and neighbouring sites, AFPhyloMix uses a Bayesian inference model to compute the phylogenetic tree and the haplotype relative abundances. Our results show that AFPhyloMix works well on both the simulated data set and the real data set.

This is a *PLOS Computational Biology* Methods paper.

## Introduction

Molecular phylogenetic reconstruction is the mainstay of modern evolutionary biology [1, 2]. To use a particularly relevant and recent example, tracing the spread of the COVID-19 pandemic, and understanding the emergence of new variants, has required the use of reliably constructed phylogenies of SARS-CoV-2 genomes [3]. DNA sequencing is used to produce the data from which such valuable phylogenies can be inferred. However, because modern sequencing technologies can produce several gigabases of nucleotide sequences in a single day, one of the challenges for the molecular phylogeneticist is to deal with this quantity of data in a timely manner while still reconstructing accurate phylogenies. To this end, phylogeneticists have developed rapid alignment and tree reconstruction algorithms [4, 5], using pre-processed and curated sequences. Pre-processing and sequence curation can be laborious, but are necessary tasks because a great deal of sequence data are generated using next generation short-read sequencing technologies. Such sequences are often barcoded using unique DNA identifier tags, and then collectively pooled and sequenced in a single run. The unique barcode allows sequences belonging to different samples to be separated computationally, before additional error-correction and subsequent down-stream analyses are performed.

Quite apart from the costs incurred by data pre-processing and curation, the preparation of barcoded sequence libraries is itself costly. More importantly, there are some samples where barcoding is impractical. For instances, rapidly evolving viruses (e.g., Human Immunodeficiency Virus (HIV) and Hepatitis C Virus (HCV)) typically exist as a collection of genetically diverse genomes within an infected individual. In some instances, hosts are infected with different strains of the same virus (i.e., coinfection). To sequence one or more target genes from a collection of these viruses using barcoding, one would need to isolate individual viral genomes before library preparation. This can be done, but again, is time-consuming and laborious.

In this paper, we describe a novel approach, AFPhyloMix (Assembly-Free Phylogenetics for Mixtures), to reconstruct the phylogeny of non-barcoded amplicons in a mixture that has been sequenced using short-read sequencing. More precisely, the input sample consists of mixtures of anonymous (i.e., non-barcoded) amplicons of a targeted locus, obtained from multiple individuals, each amplicon being longer than the read-length of sequenced fragments. We assume that all short-reads can be aligned to the same reference sequence. We have developed our method to work on samples drawn from a population of closely related individuals (i.e., from individuals within a species). In any mixture of individuals drawn from such populations, some amplicons may be identical to others. We refer to a group of identical amplicons as a haplotype [6]. The mixture, therefore, contains a collection of haplotypes, each haplotype being represented by a relative abundance between 0 and 1 (non-inclusive). AFPhyloMix estimates the phylogeny of the haplotypes and their relative abundances, without the need for reconstructing the haplotype sequences. To validate our approach, we evaluate the efficiency of the method on simulated and real data, and we discuss the conditions under which the method performs well, and its limitations.

## Methods

### Overview

The algorithm, AFPhyloMix, proceeds as follows. Given a mixture of $n$ haplotypes, with relative abundances $(r_1, r_2, \ldots, r_n)$, short-read sequencing generates $k$ sequences that can be aligned to a reference sequence. AFPhyloMix then identifies the potential sites with single-nucleotide-polymorphisms (SNPs) from this alignment of reads. Under an infinite-sites model of evolution [7], where each mutation occurs at a new site and any given SNP can have a maximum of two nucleotides, we distinguish between the frequency of a given nucleotide at a given SNP, and the number of SNPs with the same frequency distribution of nucleotides. We refer to these two types of frequencies as the *SNP ratio* and the *SNP frequency*, respectively. For example, assume that in an alignment with three SNPs, site $i$ has nucleotides $A$ and $G$ with ratios 0.75 and 0.25, respectively; site $j$ has nucleotides $C$ and $T$, with ratios 0.6 and 0.4, respectively; and site $k$ has nucleotides $G$ and $T$ with ratios 0.75 and 0.25, respectively. We will adopt the convention of using the smaller nucleotide ratios when identifying the value of a SNP ratio. Therefore, the SNP ratio for site $i$ and $k$ is 0.25 and site $j$ is 0.4. Sites $i$ and $k$ have the same frequency distribution of nucleotides, even though they may have different constituent nucleotides. In this case, the SNP frequency for the nucleotide distribution instantiated in sites $i$ and $k$ is 0.67 or 2/3. (We note that the SNP ratios and frequencies are related to the Site Frequency Spectrum [8]; however, because coverage of short-reads vary across the alignment, nucleotide ratios at each SNP vary as a continuous rational variable rather than as an integer).

In AFPhyloMix, a likelihood function computes the probability of observing the distributions of SNP ratios (data, $D$) along the alignment given their expected distributions, which is itself conditional on a specified tree topology, haplotype relative abundances, and sequencing error, assuming an infinite-sites model of evolution. A Bayesian approach is used to compute the posterior probability $P(R, T, e|D)$ of a set of parameters: the relative abundance of haplotypes ($R$), the tree topology ($T$), and the sequencing error ($e$), given the observed pattern of the data ($D$), as follows.

$$P(R, T, e|D) \sim L(D|R, T, e)P(R)P(T)P(e)$$

$L(D|R, T, e)$ is the likelihood of the observed pattern of SNPs given the relative abundances of haplotypes, tree topology, and the sequencing error. $P(R)$, $P(T)$, and $P(e)$ are the prior probabilities of the relative abundances of haplotypes, tree topology, and the sequencing error, respectively. A Bayesian Metropolis-coupled Markov chain Monte Carlo (MCMCMC) inference engine is implemented, to deliver the joint posterior probability distribution of tree topologies and haplotype relative abundances. After the Bayesian computation, the edge lengths on the tree are computed according to the SNP frequencies and the tree topology with the highest posterior probabilities.

To illustrate this approach, consider Fig 1 which shows the relationship between observed and expected SNP ratios and frequencies, along a specified tree. Given a 5-tip tree with tip relative abundances (i.e. the relative abundances of the corresponding haplotypes represented by the tips) shown in Fig 1A, a mutation $x \in \{A, C, G, T\}$ that occurs on the edge XA over evolutionary time would lead to a different nucleotide on a SNP site in haplotype A relative to other haplotypes. The expected SNP ratio of any mutation along the edge XA would be 0.075, which is the relative abundance of tip A. The number of SNPs with this mutational pattern—the SNP frequency—would depend on the length of the edge XA. Fig 1B shows the expected SNP ratio and the expected SNP frequencies (i.e. the expected ratio of the occurrences for the mutations on different edges of the tree). For example, the high expected SNP frequency of sites with SNP ratio of 0.485 is due to the mutation on the long edge XZ.

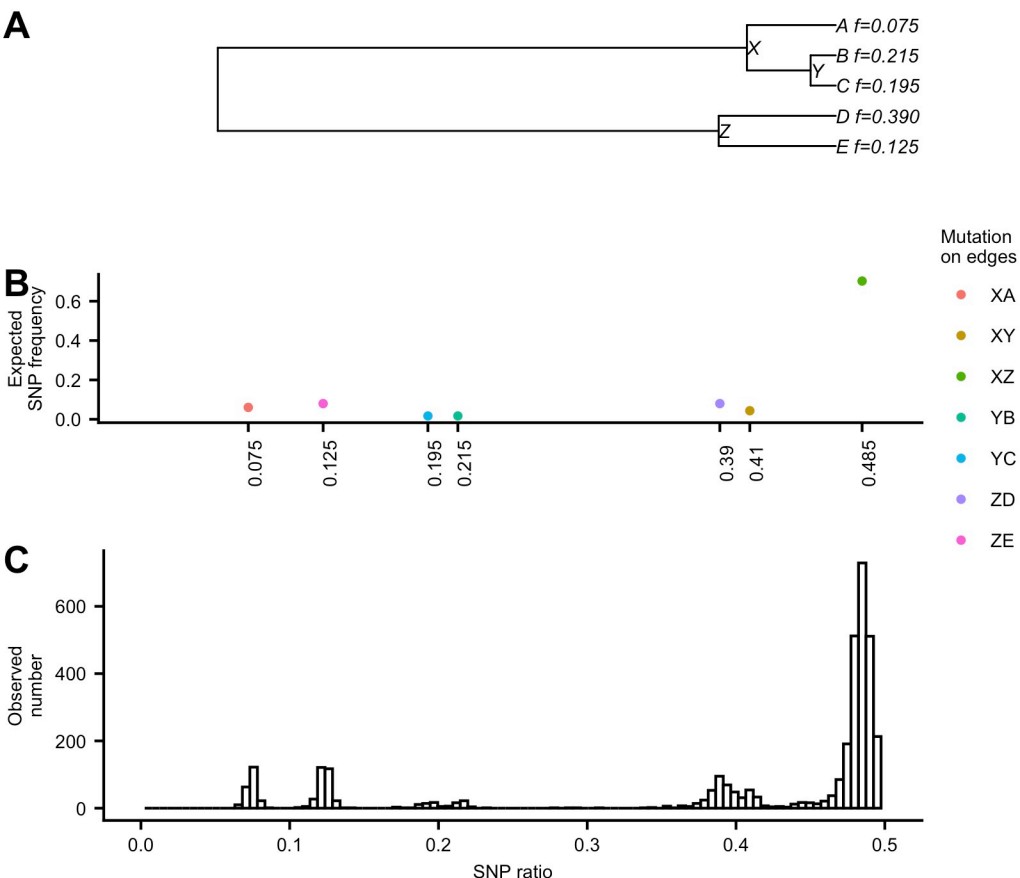

**Fig 1. SNP frequencies vs SNP ratios.** (A) An example of a 5-tip tree with expected tip relative abundances (i.e. haplotype relative abundances). (B) The expected SNP frequencies against the expected SNP ratio of the occurrences for the mutations on different edges of the tree. (C) The observed distribution of SNP ratios from the short read sequences generated from five simulated genomic sequences with expected relative abundances based on the tree in (A). To obtain observed frequencies and relative abundances, paired-end error-free reads of length 150bp with total coverage of 15,000x were simulated by ART [9] from five 50k-long haplotype sequences generated by INDELible [10] with JC model [11].

## Consideration of the connection between two SNP sites

In Fig 1B, every SNP site is treated independently. The fact that reads cover multiple SNP sites means that there is an association amongst the observed ratios for multiple SNPs; we found that modelling this association improved the accuracy of the estimation on the tree topology and the tip relative abundances. Where there is no sequencing error, as illustrated in Fig 2A, two SNP sites can generate three different combinations (*patterns of nucleotides*) on the nucleotide sequences if the mutations of the two SNP sites occur on different edges of a tree, while there are only two *patterns of nucleotides* if their mutations happen on the same edge of the tree. For example, two SNP sites with one mutation on the edge ZE and another on the edge XY, as shown in Fig 2A, lead to three different *patterns of nucleotides* on these two SNP sites with expected ratios 0.125, 0.41, and 0.465. Different locations of the mutations on the tree can result in different sets of expected ratios (Fig 2B). Similar to the compatibility problem of two sets of binary characters [12], since the infinite site model only allows one mutation along the tree for every SNP site, two SNP sites can create either two or three, but not four *patterns of nucleotides*. Moreover, how often these *patterns of nucleotides* happen depends on the product of the lengths of the edges on which the two SNP sites occur.

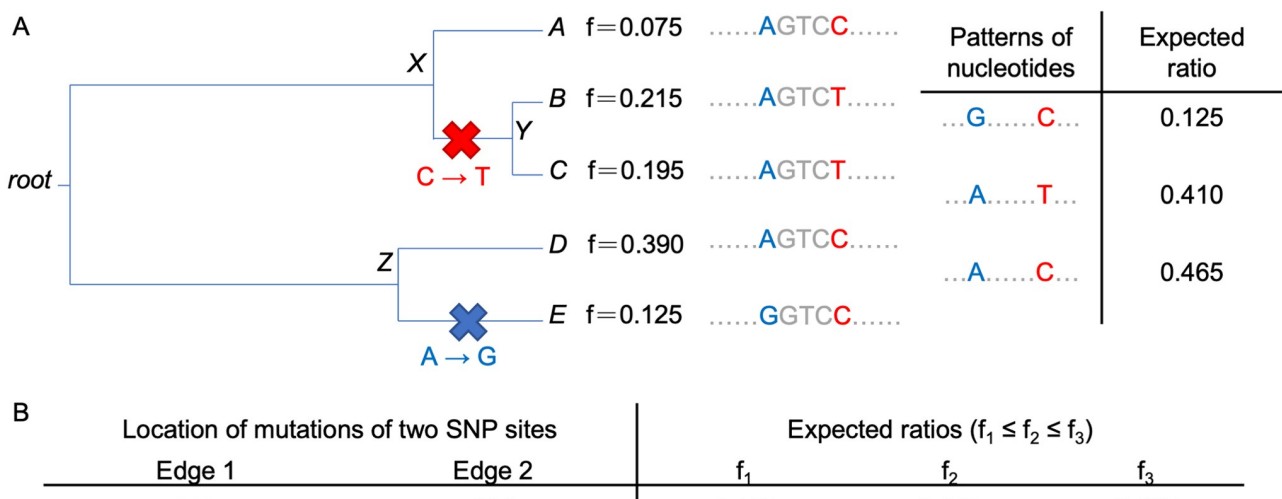

**Fig 2. Consideration of connections between two SNP sites.** (A) Under the infinite site model, by allowing one mutation along the tree for every SNP site, two SNP sites may make three different *patterns of nucleotides*. (B) Different locations of the mutations on the tree can result in different set of expected ratios.

Considering the possible three *patterns of nucleotides* (with ratios $f_1, f_2, f_3 = 1 - f_1 - f_2$ where $f_1 \leq f_2 \leq f_3$) due to the mutations of two SNP sites on different edges of the tree, Fig 3A shows the distribution of all possible pairs of $f_1$ and $f_2$ according to the tree in Fig 2A. The size of the circle represents the expected probability of occurrence. For example, the largest circle at ($f_1 = 0.125$, $f_2 = 0.39$) refers to the *patterns of nucleotides* created by two SNP sites with mutations on the edges XZ and ZE, and the probability is relatively high because of their long edge lengths.

AFPhyloMix implements a Bayesian inference methodology to estimate the posterior probability distribution of tree topologies and tip relative abundances given the observed ratios of the *patterns of nucleotides* created by pairs of SNP sites.

## Estimation of tree topology and tip relative abundances

Assume that there are $n$ haplotypes. If there is no sequencing error, there should be only two types of nucleotides on each SNP site; for convenience, we will refer to the two allowable states at a given SNP location canonically as '0' and '1'. Considering two sites $i$ and $j$, let $s(ij)$ be the nucleotides of the same read covering the sites $i$ and $j$. Also let the states of the root of the tree be $r_i$ and $r_j$, where $r_i, r_j \in \{0, 1\}$, on the site $i$ and the site $j$, respectively. Given a $n$-tip rooted tree topology $T$, a set of $n$ tip relative abundances $R$, and the edges of $T$: $\{e_1, \cdots, e_{2n-2}\}$, let $\tilde{P}(s(ij) = pq|u, v, r_i, r_j)$, where $p, q \in \{0, 1\}$ and $u, v \in \{\varepsilon, e_1, \cdots, e_{2n-2}\}$ (the empty string $\varepsilon$ represents no mutation on the site), be the expected probability of the same sequence having nucleotide $p$ on the site $i$ and nucleotide $q$ on the site $j$ given the mutations of the site $i$ and $j$ are on the edge $u$ and the edge $v$, and the states of the root of the tree are $r_i$ and $r_j$. For example, for the topology and tip relative abundances in Fig 2, when $u = ZE$, $v = XY$, $r_i = r_j = 0$,

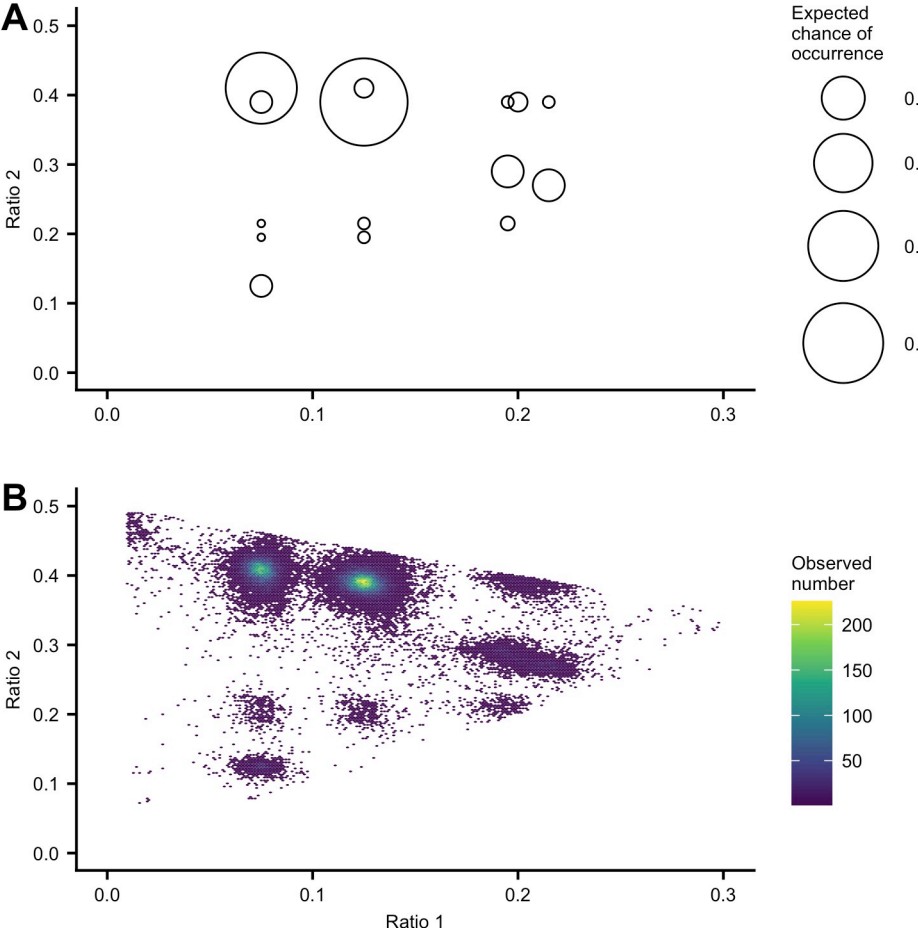

**Fig 3. Distribution of the first and the second ratios of three *patterns of nucleotides* created by two SNP sites.**
Considering the tree in Fig 2A, two SNP sites having mutations on different pair of edges can lead to three *patterns of nucleotides* with ratios $f_1, f_2, f_3 = 1 - f_1 - f_2$, where $f_1 \leq f_2 \leq f_3$. (A) The distribution of all possible pairs of $f_1$ and $f_2$. The size of the circle represents the expected chance of occurrence. (B) The distribution of pairs of observed values of $f_1$ and $f_2$ obtained from the short read sequences generated from five simulated genomic sequences referenced in Fig 1C. Every pair of SNP sites close enough to be covered by the same short reads were checked. Considering the possible three patterns of nucleotides, the pairs of observed values of $f_1$ and $f_2$ were obtained based on the set of short reads covering the pairs of SNP sites. The observed distribution is consistent with the expected distribution plotted in (A).

$\tilde{P}(s(ij) = 00|u, v, r_i, r_j) = 0.465$, $\tilde{P}(s(ij) = 01|u, v, r_i, r_j) = 0.410$,

$\tilde{P}(s(ij) = 10|u, v, r_i, r_j) = 0.125$, and $\tilde{P}(s(ij) = 11|u, v, r_i, r_j) = 0.0$.

Ideally, if there is no sequencing error, the number of combinations between the nucleotides of the reads covering the sites $i$ and $j$ should either be one (if $u = v = \varepsilon$), two (for example, when $u = v \neq \varepsilon$, or $v \neq u = \varepsilon$, or $u \neq v = \varepsilon$), or three. However, in a data set with sequencing error, the number of combinations observed may well be more (up to a maximum of 16). We will compute the expected probabilities taking account of sequencing errors. With the sequencing errors, each SNP site may contain 4 nucleotide types, say 0, 1, 2, and 3. Without loss of generality, we assume 0 and 1 are the major characters, while 2 and 3 are the characters created by the sequencing errors. Given a tree topology $T$, a set of tip relative abundances $R$, and a sequencing error rate $e$, define $P(s(ij) = pq|u, v, r_i, r_j)$, where $p, q \in \{0, 1, 2, 3\}$, as the expected probability of observing the same read having nucleotide $p$ on the site $i$ and nucleotide $q$ on the site $j$, when the mutations of the sites $i$ and $j$ are on the edges $u$ and $v$, and the

states of the root of the tree are $r_i$ and $r_j$, respectively.

$$P(s(ij) = pq|u, v, r_i, r_j) = \sum_{p'q' \in \{0,1\}} \tilde{P}(s(ij) = p'q'|u, v, r_i, r_j)\psi(p' \to p)\psi(q' \to q) \quad (1)$$

The function $\psi(p' \to p)$ where $p' \in \{0, 1\}$ and $p \in \{0, 1, 2, 3\}$ denotes the probability of observing a nucleotide $p$ on the read when the actual nucleotide should be $p'$.

$$\psi(p' \to p) = \begin{cases} 1 - e, & \text{if } p' = p \\ e/3, & \text{otherwise} \end{cases} \quad (2)$$

Again, consider two sites $i$ and $j$, and let $n_{ij}(p, q)$, where $p, q \in \{0, 1, 2, 3\}$ be the number of reads observed having nucleotide $p$ on site $i$ and nucleotide $q$ on site $j$ in the data set. Let $A_{ij} = \{n_{ij}(p, q)|p, q \in \{0, 1, 2, 3\}\}$ be the observed combinations of characters on the reads covering the site $i$ and the site $j$. Given a $n$-tip rooted tree topology $T$, a set of relative abundances of $n$ tips $R$, and a sequencing error rate $e$, define $L(A_{ij}|u, v, r_i, r_j, T, R, e)$ as the likelihood function of the alignment with sites $i$ and $j$, where $i \neq j$, provided that the mutations of the SNP sites $i$ and $j$ are on the edges $u$ and $v$, and the states of the root of the tree on the SNP sites $i$ and $j$ are $r_i$ and $r_j$, respectively. We assume that the ratios of the reads having different *patterns of nucleotides* for the sites $i$ and $j$ follow the multinomial distribution.

$$L(A_{ij}|u, v, r_i, r_j, T, R, e) \quad = \frac{\hat{n}!}{\prod_{p,q} n_{ij}(p, q)!} \prod_{p,q} P(s(ij) = pq|u, v, r_i, r_j)^{n_{ij}(p,q)}$$
$$\text{where } \hat{n} = \sum_{p,q} n_{ij}(p, q) \quad (3)$$

Practically, when performing an analysis on the alignment of the reads, for each site of the alignment, AFPhyloMix assigns the nucleotide supported by the largest number of reads to 0, the second largest to 1, the third largest to 2, and the one with the least supports to 3. There are three reasons to observe two or more nucleotides at a site:

- A site truly has a single mutational event only in its evolutionary history, but sequencing errors and other technical artifacts can introduce more than two nucleotides in the alignment of short-reads;

- A site is truly invariable over the evolutionary tree, but sequencing errors/artifacts introduce more than a single observed nucleotide in the alignment at that site;

- A site truly has experienced multiple mutational events in its events in its evolutionary history (and thus, violates the assumption of an infinite sites model).

We will address the second and third of these cases later, but if a site truly has only a single mutational event in its history, then nucleotides 0 and 1 should dominate, while nucleotides 2 and 3 will be due to sequencing errors.

Let the $n$ SNP sites be $\{S_1, S_2, S_3, S_4, \cdots, S_{n-1}, S_n\}$. One approach is to consider the patterns observed with pairs of adjacent SNP sites [i.e., if $n$ is even, then consider $(S_1, S_2)(S_3, S_4), \cdots, (S_{n-1}, S_n)$]. This approach allows, at most, only $n/2$ pairs of SNP sites to be considered. On the other hand, if we nominate a reference site, and pair each non-reference site with the reference, we can use $\approx n$ pairs of SNP sites. We have used this approach, as follows: the whole alignment is partitioned into $m$ non-overlapping windows $(W_1, W_2, \cdots, W_k, \cdots, W_m)$ of size $d$ ($d$ was set to 100 in our implementation). In each window $W_k$ a reference position $c_k \in W_k$ is selected. Let the average of the read coverage along the alignment be $cov_{avg}$. The reference site

is selected arbitrarily among those sites covered by at least $\max\{50, r^* cov_{avg}\}$ reads (where $r$ was set to 0.2). Thus, the selected reference sites will have reasonably high levels of support. For every site $i$ inside the window $W_k$, its association with the reference position $c_k$ is considered. This approach allows us to consider $n - m$ pairs of SNP sites. If such reference positions cannot be found (because the coverage of the whole window is not high enough), AFPhyloMix selects a reference covered by the highest number of reads in the window.

The likelihood of the whole alignment ($A$) given the tree topology ($T$), the tip relative abundances ($R$) and the sequencing error rate ($e$) is:

$$
\begin{aligned}
L(A|T, R, e) &= \prod_k L(W_k|T, R, e) \\
&= \prod_k \left( \sum_{r_{c_k}, v} L(W_k|r_{c_k}, v, T, R, e) Pr(r_{c_k}|T, R, e) Pr(v|T, R, e) \right)
\end{aligned}
\tag{4}
$$

The patterns obtained from the pair of site $i$ and the reference $c_k$ depends on the tree topology, the tip relative abundances, the sequencing error rate, the root states, and the edges on which the mutations occur for both the site $i$ and the reference $c_k$. Amongst all sites paired with the reference site, pattern ratios are independent after conditioning on the tree topology, the tip relative abundances, the sequencing error rate, the root state and the edge on which the mutation occurs for the reference $c_k$.

Therefore, the likelihood of the window ($W_k$) given the tree topology ($T$), the tip relative abundances ($R$), the sequencing error ($e$), the root state ($r_{c_k}$) and the edge on which the mutation occurs ($v$) for the reference $c_k$ is:

$$
\begin{aligned}
L(W_k|r_{c_k}, v, T, R, e) &= \prod_{i \in (W_k - \{c_k\})} L(A_{ic_k}|r_{c_k}, v, T, R, e) \\
&= \prod_{i \in (W_k - \{c_k\})} \left( \sum_{r_i, u} L(A_{ic_k}|u, v, r_i, r_{c_k}, T, R, e) Pr(r_i|T, R, e) \right. \\
&\qquad\qquad\qquad \left. Pr(u|T, R, e) \right)
\end{aligned}
\tag{5}
$$

Using this approach, the likelihood essentially integrates across all observed patterns, thus negating the need to identify observed nucleotide ratios that are the consequence of sequencing errors.

By substituting $L(W_k|r_{c_k}, v, T, R, e)$ from Eq 5 into Eq 4, and setting the probability of the character $r_i$ on position $i$ ($Pr(r_i|T, R, e)$) equal to the observed proportion of the nucleotide $r_i$ in the alignment, the likelihood of the alignment ($A$) given the tree topology ($T$), the tip relative abundances ($R$) and the sequencing error rate ($e$) becomes:

$$
\begin{aligned}
L(A|T, R, e) &= \prod_k \left( \sum_{r_{c_k}, v} \left( \prod_{i \in (W_k - \{c_k\})} \left( \sum_{r_i, u} L(A_{ic_k}|u, v, r_i, r_{c_k}, T, R, e) Pr(r_i) \right. \right. \right. \\
&\qquad\qquad \left. \left. \left. Pr(u|T, R, e) \right) \right) Pr(r_{c_k}) Pr(v|T, R, e) \right)
\end{aligned}
\tag{6}
$$

where $Pr(r_i) = $ the observed proportion of the nucleotide $r_i$ in $A$

$Pr(u|T, R, e)$, which is the probability of having a mutation on the edge $u$ given $T$, $F$, and $e$, can be approximated as the length of the edge $u$.

The following approximation has fewer parameters and allows MCMC to converge at a faster rate:

$$L(A|T, R, e) \approx \prod_k \Big( \max_v \Big( \sum_{r_{c_k}} \Big( \prod_{i \in (W_k - \{c_k\})} \Big( \max_u \Big( \sum_{r_i} L(A_{ic_k}|u, v, r_i, r_{c_k}, T, R, e) Pr(r_i) \Big) \Big) \Big) Pr(r_{c_k}) \Big) \Big) \tag{7}$$

For each site, this equation considers all possible edges $u$ (and $v$) on which the mutation occurs and assumes that the edge the mutation occurs on should have the highest likelihood value. This equation does not contain the parameters of $Pr(u|T, R, e)$, and is defined only by the tree topology ($T$), the tip relative abundances ($R$) and the error rate ($e$). This equation allows MCMC to converge much faster. After the estimation on the tree topology, tip frequencies and error rate, the edge lengths will be computed in the next step. The details will be shown in the later subsection "Estimation on edge lengths".

We now define $P(T, R, e|A)$ as the posterior probability of the tree topology $T$, the tip relative abundances $R$, and the sequencing error $e$ given the alignment $A$.

$$P(T, R, e|A) \propto L(A|T, R, e)P(T)P(R)P(e)$$
$$\text{where } P(T), P(R), P(e) \text{ are the prior probabilities of } T, R, e, \text{ respectively} \tag{8}$$

## Identifying invariable sites

As noted above, a truly invariable site may appear to be a SNP because of sequencing errors. To speed up the computation, AFPhyloMix skips all the sites which are identified as invariable sites. Let the maximum value of the sequencing error rate be $e_{max}$ (in our application of AFPhyloMix, $e_{max}$ is set to 0.01). A site is regarded as an invariable site if there exists only one nucleotide such that the percentage of the reads having that nucleotide covering the site is higher than $e_{max}$. Let $I$ be the set of these sites which are identified as invariable sites; then

$$L(A|T, R, e) \propto \prod_k \Big( \max_v \Big( \sum_{r_{c_k}} \Big( \prod_{i \in (W_k - \{c_k\} - I)} \Big( \max_u \Big( \sum_{r_i} L(A_{ic_k}|u, v, r_i, r_{c_k}, T, R, e) Pr(r_i) \Big) \Big) \Big) Pr(r_{c_k}) \Big) \Big) \tag{9}$$

## Markov chain Monte Carlo implementation

AFPhyloMix adopts a Bayesian model of inference to obtain an estimate of the joint posterior probability of phylogenies, haplotype relative abundances, and sequencing error, using Markov chain Monte Carlo (MCMC) sampling [13, 14]. The MCMC approach has been extensively used in phylogenetic analysis [15, 16], but sampling chains may not mix as well as they should. To overcome this, the Metropolis-coupled Markov chain Monte Carlo (MCMCMC) approach was developed by [17]. Although MCMCMC requires multiple parallel sampling chains to be run simultaneously (and thus, has demanding computational overheads), the approach has been demonstrated to improve mixing and convergence to a stationary posterior probability distribution [18].

We implemented MCMCMC in AFPhyloMix, and the details of implementation are listed in the Supporting information. AFPhyloMix reports the posterior probabilities, the tree topology, and the tip relative abundances along the cold chain every, by default, 100 rounds. Because the tip relative abundances may change along the chain, the consensus of the trees is

not easy to compute. For the sake of simplicity, AFPhyloMix also reports the tree topology and the tip relative abundances which gives the highest value among all the resulting posterior probabilities along with the computation. The performance of AFPhyloMix was then evaluated based on this tree topology and the tip relative abundances.

## Estimation on edge lengths

After AFPhyloMix estimates the topology $T$, the tip relative abundances $R$, and the sequencing error $e$ for the mixture of short read sequences from $n$ haplotypes, AFPhyloMix calculates the edge lengths in $T$ (with edges $e_1, \cdots, e_{2n-2}$) by the following method.

Let $length(u)$ be the length of edge $e_u$. In AFPhyloMix, $length(u)$ is approximated as the probability of having mutation on edge $u$ along the tree. Note that $length(u = 0)$ is the probability of no mutation along the tree (i.e. $\Sigma_{0 \le u \le 2n-2} \, length(u) = 1$).

$$
\begin{aligned}
length(u) \quad &= Pr(\text{edge } u) \\
&= \frac{\sum_i Pr(\text{edge } u \text{ at site } i)}{\sum_{u'} \{\sum_i Pr(\text{edge } u' \text{ at site } i)\}}
\end{aligned}
$$

where $Pr(\text{edge } u \text{ at site } i)$ is the probability of a mutation on edge $u$ of the tree at site $i$.

As described above, the whole genome is partitioned into non-overlapping windows $(W_1, W_2, \cdots, W_k, \cdots)$ of size $d$ ($d$ was set to 100), and inside each window $W_k$ a reference site $c_k \in W_k$ is selected. For every site $i$ (say it is inside the window $W_k$), we consider the connection between the site $i$ and the reference position $c_k$. For $i \ne c_k$, let $n_{ic_k}(p, q)$, where $p, q \in \{0, 1, 2, 3\}$, be the number of reads observed having nucleotide $p$ at site $i$ and nucleotide $q$ at site $c_k$ on the same read. Define $A_{ic_k} = \{n_{ic_k}(p, q) | p, q \in \{0, 1, 2, 3\}\}$ as the observed combinations of characters on the same reads at the sites $i$ and $c_k$. Similarly, for $i = c_k$, let $n_{c_k}(q)$, where $q \in \{0, 1, 2, 3\}$ be the number of reads observed having nucleotide $q$ at site $c_k$, let $A_{c_k} = \{n_{c_k}(q) | q \in \{0, 1, 2, 3\}\}$ be the observed pattern of characters on the reads at the site $c_k$, and let $\hat{T}, \hat{R}, \hat{e}$ be the estimated values of topology, tip relative abundances, and sequencing error, respectively.

To calculate $Pr(\text{edge } u \text{ at site } i)$, two cases are considered:

- Case 1: $i \ne c_k$ (i.e. the site $i$ is not the reference site of the window);

- Case 2: $i = c_k$ (i.e. the site $i$ is exactly the reference site of the window).

For case 1,

$$
Pr(\text{edge } u \text{ at site } i)
$$

$$
= \frac{\sum_v \left\{ \sum_{r_{c_k}} \left\{ \sum_{r_i} L(A_{ic_k} | u, v, r_i, r_{c_k}, \hat{T}, \hat{R}, \hat{e}) Pr(r_i) \right\} Pr(r_{c_k}) \right\} Pr(\text{edge } v \text{ at site } c_k)}{\sum_{u'} \left\{ \sum_v \left\{ \sum_{r_{c_k}} \left\{ \sum_{r_i} L(A_{ic_k} | u', v, r_i, r_{c_k}, \hat{T}, \hat{R}, \hat{e}) Pr(r_i) \right\} Pr(r_{c_k}) \right\} Pr(\text{edge } v \text{ at site } c_k) \right\}}
$$

For case 2,

$$
Pr(\text{edge } u \text{ at site } c_k) = \frac{\sum_{r_{c_k}} L(A_{c_k} | u, r_{c_k}, \hat{T}, \hat{R}, \hat{e}) Pr(r_{c_k})}{\sum_{u'} \left\{ \sum_{r_{c_k}} L(A_{c_k} | u', r_{c_k}, \hat{T}, \hat{R}, \hat{e}) Pr(r_{c_k}) \right\}}
$$

$Pr(r_i)$ and $Pr(r_{c_k})$ are the probabilities of the root states $r_i$ and $r_{c_k}$, which are set to the observed proportions of $r_i$ and $r_{c_k}$ in $A$, respectively. $L(A_{ic_k} | u, v, r_i, r_{c_k}, \hat{T}, \hat{R}, \hat{e})$ is the likelihood value of the observed combinations of characters on the same reads at the sites $i$ and $c_k$ given the estimated topology $\hat{T}$, the estimated tip relative abundances $\hat{R}$, the estimated sequencing

error $\hat{e}$, the mutations on edges $u$ and $v$, and the root states $r_i$ and $r_{c_k}$ for the sites $i$ and $c_k$, respectively, while $L(A_{c_k}|u, r_{c_k}, \hat{T}, \hat{R}, \hat{e})$ is the likelihood value of the observed pattern of characters on the reads at the site $c_k$ given $\hat{T}, \hat{R}, \hat{e}$, the mutation on edge $u$ and the root state $r_{c_k}$ for the site $c_k$. The computation of $L(A_{ic_k}|u, v, r_i, r_{c_k}, \hat{T}, \hat{R}, \hat{e})$ is listed in Eq 3, while the calculation of $L(A_{c_k}|u, r_{c_k}, \hat{T}, \hat{R}, \hat{e})$ can be done similarly. Whereas it is theoretically possible to simultaneously infer edge lengths and topologies, in preliminary simulations, we have found that this does not deliver accurate results. We think that this is because, in the absence of sequence information at the tips, inferred mutations will naturally favour long branches, thus lowering the probability of seeing short branches. Instead, we adopt a two-step approach, in which the first step focuses on optimization of the tree topology, the tip relative abundances, and the sequencing error rate, while the second step computes the values of the edge lengths. Because of the fewer number of parameters needed to be estimated in the first step, we see that the MCMC chain converges at a faster rate, especially as the number of haplotypes increases.

## Removal of the sites that violate the infinite sites model

As noted previously, AFPhyloMix uses an infinite sites model of evolution, and thus assumes that every site has no or only one mutation in its evolutionary history. Before AFPhyloMix proceeds to estimate the topology and the tip relative abundances, AFPhyloMix examines the read alignments and attempts to identify SNP sites which have more than one mutation. The procedure to identify these sites is as follows:

1. If the SNP site has more than two different nucleotides supported by at least $e_{max}$ (i.e. 1%) of the reads, then the SNP site is ignored.

2. We consider SNP sites with only two different nucleotides, with each nucleotide supported by at least 1% of the reads. If there is no back/hidden mutation, two SNP positions will give at most three combinations (as mentioned before). Based on this observation, the following simple method has been developed to identify the sites which are likely to have back/hidden mutations:
   For a SNP site $i$, we check all other SNP sites $j$ such that there are sufficient number of reads covering both sites $i$ and $j$. If there are at least $d$ sites (i.e. $j_1, j_2, \cdots, j_d$) which separately have more than three combinations supported by at least 1% of the reads when paired with site $i$, then site $i$ is regarded having back/hidden mutations and it is discarded. We have tested for different values of $d$ on simulated data and found that when $d = 3$ the method performed well in terms of accuracy.

## Experiment and results

Both simulated and real data were used to evaluate AFPhyloMix.

### Simulated data

To evaluate AFPhyloMix using simulated data sets, six hundred data sets were simulated and each data set was a mixture of various numbers ($n$) of haplotypes, where $n \in \{5, 7, 9, 11, 13, 15\}$ (100 data sets each). In each data set, the $n$ haplotypes were mixed in different random concentrations (with the difference between any two haplotypes $\geq 0.0033$). Paired-end reads of length 150bp with total coverage of 15,000x (where the coverage of the haplotype with the least concentration is 250x) were simulated by ART with the Illumina HiSeq 2500 error model—HS25, from $n$ 50k-long haplotype sequences, which were generated by INDELible using JC model

from a *n*-tip tree with 0.03 root-to-tip distance randomly created by Evolver [19] from PAML package [20].

The root sequence reported by INDELible [10] was used as the reference sequence. After using BWA [21] to align the reads against the reference sequence, we ran AFPhyloMix under the default settings on the read alignments to estimate the tree topology and the tip relative abundances (i.e., haplotype concentrations) for the mixture. By default, AFPhyloMix runs 8 MCMC processes in parallel: one cold chain and seven hot chains, and each chain runs 650K (for mixture with 5 sub-samples) to 1150K (for 15 sub-samples) iterations, depending on the number of edges in the resulting tree. Fig 4 shows a result from AFPhyloMix on a simulated data set with a mixture of 15 sub-samples. Fig 4A displays the posterior probabilities along the cold chain of MCMCMC process while running the AFPhyloMix. The posterior probabilities increased rapidly during the burn-in period and then appeared to stabilise to an equilibrium distribution. Fig 4B shows the distribution of tip relative abundances along the cold chain of MCMCMC process from AFPhyloMix. The distribution was found to match the expected tip relative abundances (marked with red dots) in the real tree used to simulate the data set. Fig 4C is the resulting tree from AFPhyloMix with tip relative abundances. This is the tree with the highest posterior probability along the cold chain of the MCMCMC process. The tree is topologically congruent with the real tree (Fig 4D) and the tip relative abundances are also very similar with the real sub-sample concentrations.

Fig 5A shows the summary on the accuracy of AFPhyloMix running on the simulated data sets. Among the data sets with the same number of haplotypes, the figure shows the percentage of data sets with *correct* estimation on both topologies and tip relative abundances. An estimated result is regarded as *correct* if the tips on an estimated tree can be paired up with the tips on the actual tree satisfying the following conditions: (1) the difference between the estimated tip relative abundances and the corresponding true tip relative abundances is less than 0.01; and (2) their unrooted topologies are identical. Details of the method on pairing between the tips on an estimated tree and those on the actual tree is listed in Supporting information. From the figure, AFPhyloMix achieved at least 80% accuracy for the mixtures with up to 15 haplotypes.

To further examine the deviation between the predicted tip relative abundances and the actual sample concentrations, and that between the estimated and the actual edge lengths, for each data set with *correct* estimation, we computed the root-mean-square value of the differences between the estimated and the actual values. Fig 5B and 5C show the summary on the distributions of the root-mean-square of the differences for the tip frequencies and the edge lengths. The values rise gradually as the number of haplotypes increases. For tip relative abundances, the root-mean-square values were all below 0.0012, while over 75% of the cases were below 0.0008. For edge lengths, the root-mean-square values were all below 0.005.

The pooled 95% highest posterior density of the MCMCMC estimate of error rates on these simulated data sets was between 0.00199 and 0.00204 with an average 0.00202. ART [9] reports the errors for the reads simulated by the error model HS25; in our simulations, we obtained a simulated error rate of 0.00190. AFPhyloMix relies on the alignment of reads against a root sequence to obtain the marginal posterior probability distribution of error rates. We expect, therefore, that the slightly higher value ($\approx$ +0.00012) of the MCMCMC estimate of error rates on these simulated data sets, compared with the simulated error rate reported by ART, is likely due to imperfect alignment between simulated reads and the root sequences.

The experiments were conducted on machines with 4 x 16-core Intel Xeon 2.10GHz CPUs. The running times of AFPhyloMix were listed in the Table 1.

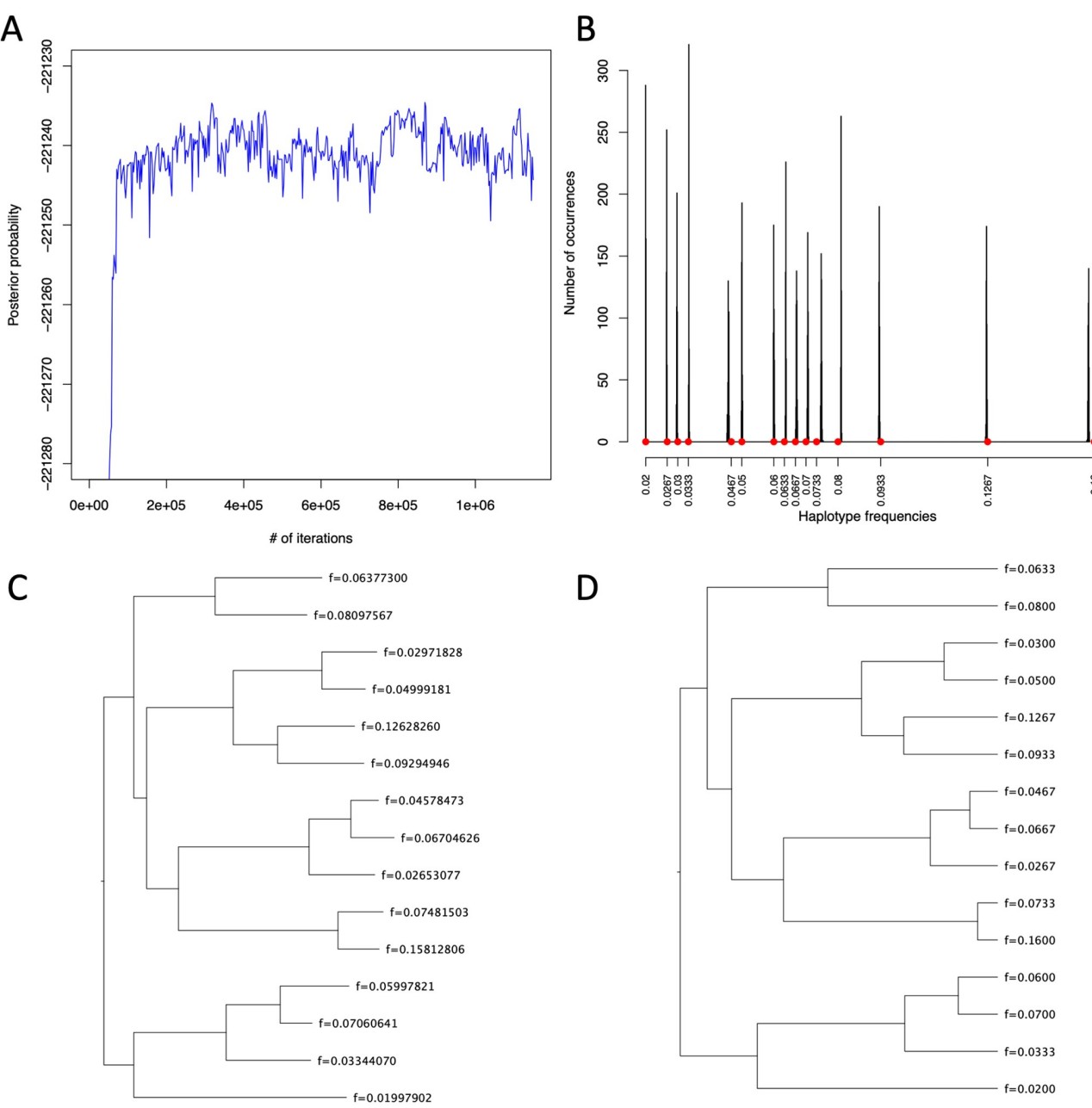

**Fig 4. The result of AFPhyloMix on a simulated data set with a mixture consisting of 15 sub-samples.** AFPhyloMix was run under the default settings on the read alignments of a simulated mixture of 15 sub-samples in order to estimate the tree topology and the tip relative abundances. (A) The posterior probabilities along the cold chain of MCMCMC process while running the AFPhyloMix, which increased rapidly during the burn-in period and then frustrated steadily over a range of values and was reaching a convergence. (B) The distribution of tip relative abundances (i.e. sub-sample concentration) along the cold chain of MCMCMC process from AFPhyloMix. The actual sub-sample concentrations are marked by the red dots. (C) The tree with tip relative abundances having highest posterior probability along the computation reported by AFPhyloMix. (D) The real tree with the actual sample concentration used for simulating the data set.

### Real data

Apart from the simulated data sets, mixtures of reads from kangaroo haplotypes were also used to evaluate the performance of AFPhyloMix.

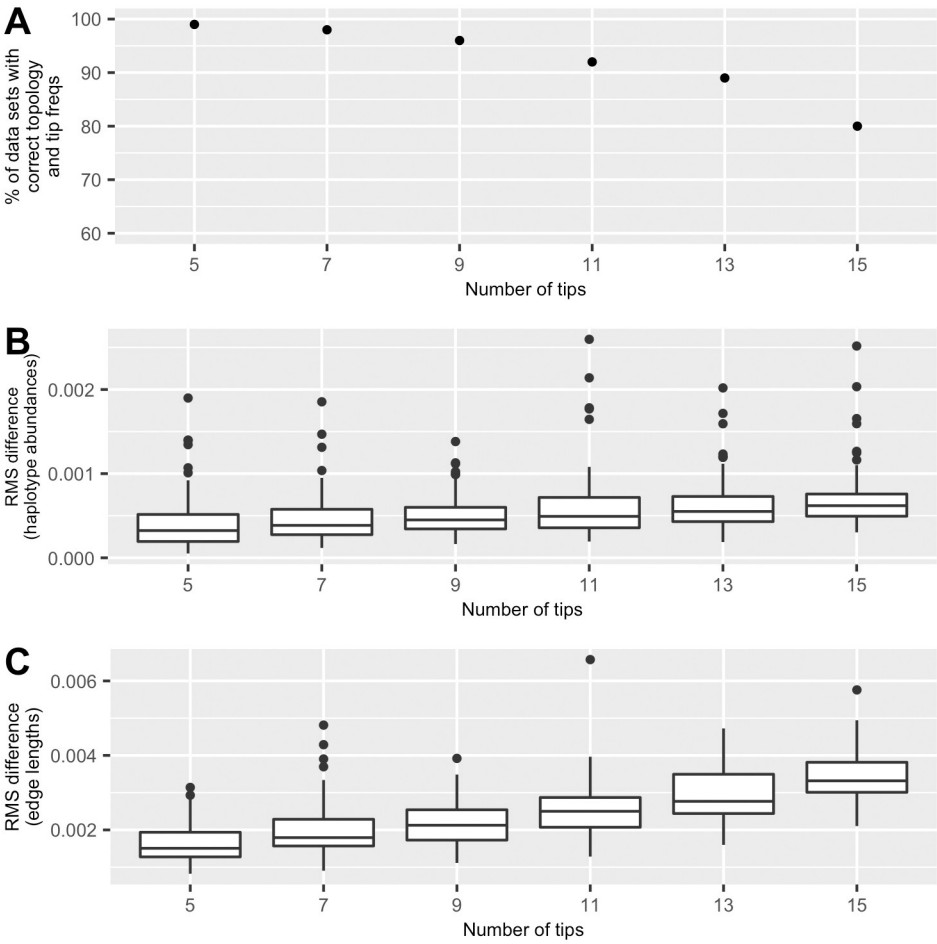

**Fig 5. Performance of AFPhyloMix on simulated data.** (A) Accuracy of AFPhyloMix—The percentage of data sets (out of 100), for different number of haplotypes, with correct estimated topologies and predicted tip relative abundances. (B) Root-mean-square differences between the actual- and the predicted tip relative abundances. (C) Root-mean-square differences between the actual and the predicated edge lengths. The accuracy decreased gradually while the variation between the estimated and the actual haplotype relative abundances and the edge lengths increased with the number of tips.

**DNA material collection and extraction.** In Australia, kangaroo are not farmed, but are culled annually to control population numbers. Culled animals are butchered by certified butchers, and the meat is sold in supermarkets. As the exact provenance of kangaroo meat sold at supermarkets is unknown, meat (produced by Macro Meats) was purchased at local super-markets in the Australian Capital Territory (Coles Supermarkets Australia Pty Ltd and

**Table 1. Average running time of AFPhyloMix on the simulated data sets.**

| Number of tips | Running time |
|---|---|
| 5 | 4.0 hours |
| 7 | 14.5 hours |
| 9 | 22.3 hours |
| 11 | 39.8 hours |
| 13 | 76.8 hours |
| 15 | 112.6 hours |

Woolworths Ltd) on 10 separate occasions over one year (from 29th May 2018 to 26th July 2019), to avoid sampling the same animal. According to statistics from the Department of the Environment and Energy of Australia, they should be of genus *Macropus*, and most likely eastern grey kangaroo, *Macropus giganteus*, as it is the largest population with the highest quota in New South Wales, Australia [22].

Approximately 30 mg of meat was excised and homogenized for each individual. Genomic DNA was then extracted using the DNeasy Blood & Tissue Kits (Qiagen) following the manufacturer's protocol.

**Amplification and sequencing.**   Long PCR amplification of complete kangaroo mitochondrial genomes was carried out using the pair of primers (Lt12cons: 5'- GGGATTAGA TACCCCACTAT -3', HtPhe: 5'-CCATCTAAGCATTTTCAGT -3'), which was selected from a previous study [23]. PCR reactions was performed using Takara PrimeSTAR GXL DNA Polymerase under the following conditions: 1 min initial denaturation at 95˚C, followed by 30 cycles of 10 s at 98˚C, 15 s at 55˚C, and 15 min at 68˚C. The PCR products were electrophoresed in 1% agarose gel, purified the fragments, and then randomly fragmented to 650 bp by sonication (Covaris S220).

Library preparation and sequencing were performed by GENEWIZ. Amplified fragments of all 10 individuals were sequenced under the same run. In order to obtain a highly reliable phylogenetic tree of these sub-samples for evaluating our method, each individual was barcoded with unique indices before multiplexing and sequencing, so that each short read sequence could be identified to the corresponding sub-sample. The relative concentrations of the sub-samples are listed in the Table 2 (2$^{nd}$ column). Sequencing was performed on an Illumina MiSeq machine with paired-end read length of 2 x 300 bp.

**Phylogenetic tree reconstruction.**   To start with, a reliable phylogenetic tree of the haplotypes was constructed, so that this gold-standard result could be used to evaluate our method. First, all the short read sequences were demultiplexed into sub-samples according to the barcodes appended on the sequences. Then *de-novo* assembly was performed on the short reads for each sub-sample separately by SOAPdenovo-Trans-127mer from SOAPdenovo-Trans package [24] with parameter: kmer-size = 91. After the mitochondria DNA sequences of the 10 haplotypes were constructed, a multiple sequences alignment was computed by MAFFT [4] with G-INS-i strategy in Geneious 11.1.5 [25]. The phylogenetic analysis was conducted using IQ-TREE [26] with the evolutionary model HKY+F+I, and the ML phylogenetic tree (shown in Fig 6B) was used as the reference tree to evaluate our method.

**Table 2. Concentration of kangaroo sub-samples.**

| Sub-sample ID | Concentration | Concentration without K01 |
|:---:|:---:|:---:|
| K01 | 0.019 | - |
| K02 | 0.029 | 0.030 |
| K03 | 0.045 | 0.046 |
| K04 | 0.047 | 0.048 |
| K05 | 0.094 | 0.096 |
| K06 | 0.102 | 0.104 |
| K07 | 0.125 | 0.127 |
| K08 | 0.130 | 0.132 |
| K09 | 0.188 | 0.192 |
| K10 | 0.221 | 0.225 |
| Total | 1.000 | 1.000 |

The relative concentration between different kangaroo sub-samples.

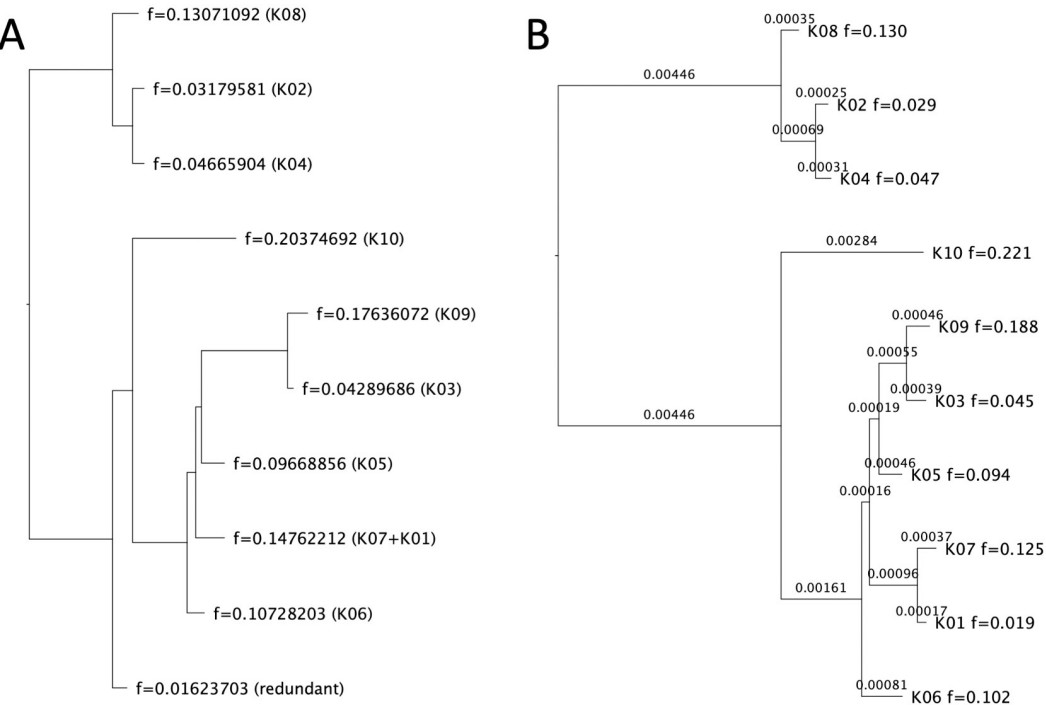

**Fig 6. Result on a real data set with mixture of 10 sub-samples.** (A) The tree with tip relative abundances reported by AFPhyloMix. (B) The tree reported by IQ-Tree [26]. Note that the tip labels inside the brackets in (A) were added manually after comparing with the tree reported by IQ-Tree, indicating the corresponding sample that each tip should be assigned to according to the topology and the tip relative abundances.

The short read sequences from 10 haplotypes were then mixed, and the barcode of each haplotype was removed. Trimmomatic [27] was run on the mixture of reads to remove adaptors, leading and trailing low quality bases by using the options: "ILLUMINACLIP:Tru-Seq3-PE-2.fa:2:30:10 LEADING:3 TRAILING:3 SLIDINGWINDOW:4:15 MINLEN:36". Next, the reads were aligned to the reference sequence of the eastern gray kangaroo mitochondrial genome (GenBank Accession Number: NC_027424) by BWA [21] and the alignments with mapping quality score (MAPQ) lower than 20 were discarded.

AFPhyloMix was used with the short-read alignment to estimate the phylogenetic tree and the relative abundances of each haplotype. For each read, AFPhyloMix discarded the nucleotides with base quality score lower than 25. Fig 6A shows the reported tree, as well as the tip relative abundances, with the highest posterior probability obtained. The tip labels inside the brackets were added manually after comparing with the tree reported by IQ-Tree (in Fig 6B), indicating the corresponding sample that each tip should be assigned according to the topology and the tip relative abundances. Overall, the topology and the tip relative abundances recovered by AFPhyloMix matched the tree from IQ-Tree, except that AFPhyloMix combined two haplotypes K01 and K07 into one. From the tree reported by IQ-TREE, the distance between the tips K01 and K07 is 0.00054 (99.946% similarity between these two sequences). The method could not distinguish the two too-similar sequences, and thus regarded them as the same sequence.

To determine whether these two nearly identical sequences affected the performance of AFPhyloMix, we removed the sub-sample of K01 and re-estimated the phylogeny with the same method described above. The updated relative concentrations of the remaining haplotypes among the mixture is shown in Table 2 (3rd column). Both AFPhyloMix and IQ-TREE

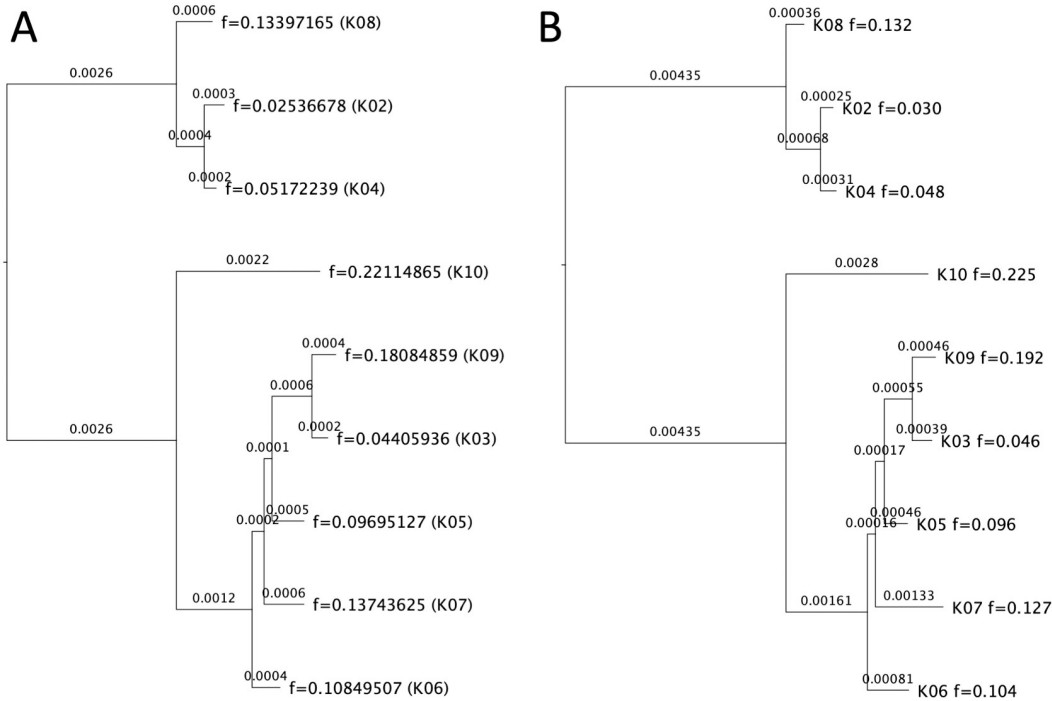

**Fig 7. Result on a real data set with mixture of 9 sub-samples.** The haplotype K01 was removed from the mixture and the experiment was repeated. (A) The tree with tip relative abundances reported by AFPhyloMix. (B) The tree reported by IQ-TREE [26]. Note that the tip labels inside the brackets in (A) were added manually after comparing with the tree reported by IQ-Tree, indicating the corresponding sample that each tip should be assigned to according to the topology and the tip relative abundances.

analyses resulted in the same topology associated with tip relative abundances well matched the concentrations of those 9 haplotypes (in Fig 7).

The 95% highest posterior density of the MCMCMC estimate of error rates on the real data sets was between 0.000722 and 0.000735 with an average 0.000729. In order to compute the underlying actual error rate on the real data set, reads which had been processed by Trimmomatic [27], were compared with the corresponding assembled haplotype and the error rate of 0.000713 was obtained, after discarding the read bases with base quality scores lower than 25 (the same criteria AFPhyloMix used to filter out the low-quality read bases). Again, the slightly higher value ($\approx$ +0.000016) of the MCMCMC estimate of error rates on the real data sets compared with the obtained error rate is consistent with what we observed with simulated data, and is likely due to the imperfect alignment between the reads and the reference sequence.

## Discussion

This research demonstrates the feasibility of reconstructing a phylogenetic tree directly from the short read sequences obtained from a mixture of closely related amplified sequences, without barcoding. The results indicate that our methods work well on the simulated data set for a mixture of reads generated from up to 15 haplotypes and on a real data set of a mixture with 10 haplotypes.

Perhaps unsurprisingly, AFPhyloMix worked better on the simulated data sets than in the real data sets when it came to estimating haplotype concentrations. The root-mean-square difference between the estimated sub-samples concentrations and the expected concentrations in the real data sets was 0.0059 (from Fig 7), which was larger than those in the simulated data

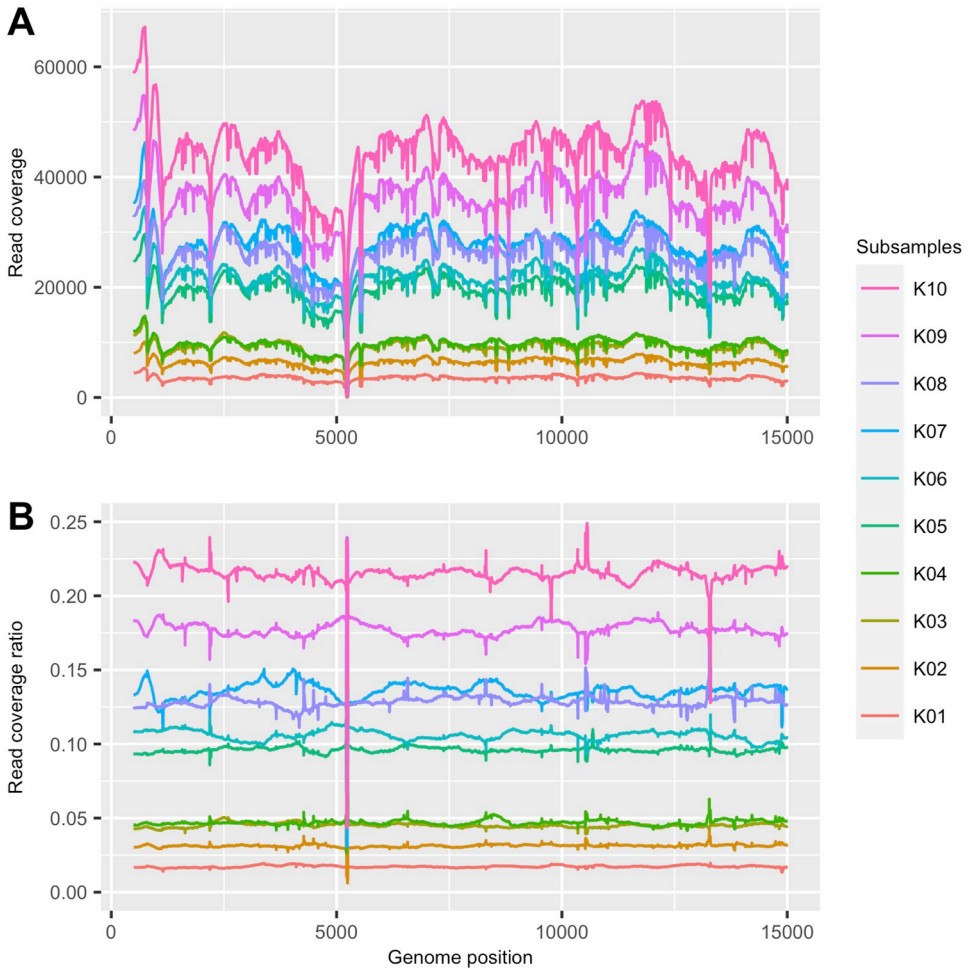

**Fig 8. Distribution of the read alignments of 10 haplotypes against the reference genome.** (A) The absolute read coverages along the genome. (B) The ratios on read coverages between sub-samples along the genome.

sets (i.e. all were below 0.0027 in Fig 5B). Of course, the expected haplotype concentrations may have differed from the true concentrations in the mixture: the physical act of mixing small volumes could have led to differences in the relative concentrations of haplotypes, and this may have contributed to a higher-than-expected root-mean-square.

Another factor affecting the performance of the method is the varying coverage of short read sequences along the genome. Fig 8A shows the actual distribution of the read alignments of 10 haplotypes along the genome. We expected read coverage to vary along the genome randomly, without any association to haplotypes. Surprisingly, we found that, when all the haplotypes were sequenced under the same run (i.e. all haplotypes were pooled into the same library before sequencing), the read coverages of the haplotypes had similar trends: all had relatively high (or low) read coverages at the same regions of the genome. The similar trends of the read coverages along the genome between the haplotypes led to a consistent distribution of the ratios of haplotypes along the genome (Fig 8B). The consistent read coverage ratios along the genome worked in our method's favor; however, we noticed that a few short regions on the genome had sudden changes in the ratios of the read coverage amongst haplotypes (Fig 8B). In these regions, some reads were trimmed after the alignment against the reference sequence or could not be aligned to the reference sequence due to the dissimilarity between the read

sequences and the reference sequence. Another preprocessing step was therefore developed in AFPhyloMix, in which read alignments were examined and problematic regions removed if more than $r$ reads in those regions were trimmed ($r$ was set to 2% of the read coverage).

AFPhyloMix's performance also depends on the SNP frequency and the read length. Too short a read length or too low a SNP frequency will mean that any given read will be unlikely to have more than a single SNP. If this happens, it will confound AFPhyloMix's use of SNP pairs.

AFPhyloMix only considers the association between two SNP sites, but it is sensible to consider the association between more SNP sites in order to acquire higher sensitivity of the methods especially when the number of haplotypes increases. The time complexity of the algorithm is $O(mn^d)$ where $n$ is the number of haplotypes, $m$ is the number of potential SNP sites, and $d$ is the number of SNP sites used to construct patterns of nucleotides. In our current implementation of AFPhyloMix, $d = 2$. The running time of the algorithm will increase exponentially as $d$ increases. It will be a challenge to come up with a faster algorithm and consider the association between more SNP sites.

Finally, it is worth noting that we have applied AFPhyloMix to sequences of closely related individuals—in our simulations, we set a root-to-tip distance of 0.03. The assumption of an infinite sites model that is applied in AFPhyloMix is appropriate for closely-related individuals. Amongst other things, the infinite sites model allows us to constrain the number of site patterns we expect to see, and use deviations from these expected patterns to error-correct. To extend our algorithm to more divergent sequences will require a different model of mutation. This remains a work in progress.

## Conclusion

AFPhyloMix is designed to estimate the concentration of haplotypes and reconstruct the phylogenetic tree directly from the short read sequences in the mixture of haplotypes with no barcode, given that the number of haplotypes is known. This research demonstrates the feasibility of our approach, and is a first attempt to infer the phylogenetic tree from the mixture of reads unidentifiable to haplotypes, bypassing the assembly process of multiple genomic sequences. The experimental results have demonstrated that AFPhyloMix works reasonably well for both simulated data and real data.

## Supporting information

**S1 Fig. Performance of AFPhyloMix on simulated data with different read coverage.** (A) Accuracy of AFPhyloMix between data sets of which the least abundant haplotype has at least 250x and 100x read coverages. (B) Root-mean-square differences between the actual and the predicted tip relative abundances for data sets with different read coverages. (C) Root-mean-square differences between the actual and the predicated edge lengths for data sets with different read coverages.
(TIF)

**S2 Fig. Performance of AFPhyloMix on simulated data evolved under two different substitution models.** (A) Accuracy of AFPhyloMix between data sets evolved under a simple model —JC and a model with site variation—JC+G. (B) Root-mean-square differences between the actual and the predicted tip relative abundances for data sets evolved under JC and JC+G. (C) Root-mean-square differences between the actual and the predicated edge lengths for data sets evolved under JC and JC+G.
(TIF)

**S1 Text. Evaluation of AFPhyloMix on other simulated data sets.**
(PDF)

**S2 Text. Implementation details in Markov chain Monte Carlo.**
(PDF)

**S3 Text. To check the correctness of an estimated tree and report the minimum root-mean-square difference between the estimated and the actual haplotype relative abundances.**
(PDF)

## Acknowledgments

The authors thank David Bryant for constructive and helpful comments on the manuscript. This work was supported by computational resources provided by the Australian Government through the Australian National University under the National Computational Merit Allocation Scheme.

## Author Contributions

**Conceptualization:** Thomas K. F. Wong, Steven H. Wu, Jeet Sukumaran, Allen G. Rodrigo.

**Data curation:** Thomas K. F. Wong, Teng Li.

**Formal analysis:** Thomas K. F. Wong.

**Funding acquisition:** Teng Li, Allen G. Rodrigo.

**Investigation:** Thomas K. F. Wong.

**Methodology:** Thomas K. F. Wong, Teng Li, Louis Ranjard, Allen G. Rodrigo.

**Project administration:** Thomas K. F. Wong, Louis Ranjard, Allen G. Rodrigo.

**Resources:** Allen G. Rodrigo.

**Software:** Thomas K. F. Wong.

**Supervision:** Allen G. Rodrigo.

**Validation:** Thomas K. F. Wong, Steven H. Wu, Jeet Sukumaran.

**Visualization:** Thomas K. F. Wong.

**Writing – original draft:** Thomas K. F. Wong.

**Writing – review & editing:** Thomas K. F. Wong, Teng Li, Louis Ranjard, Steven H. Wu, Jeet Sukumaran, Allen G. Rodrigo.

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
