## [Decision Letter · Decision Letter 0]

27 May 2021

Dear Dr Wong,

Thank you very much for submitting your manuscript "An assembly-free method of phylogeny reconstruction using short-read sequences from pooled samples without barcodes" for consideration at PLOS Computational Biology. Your manuscript was reviewed by members of the editorial board and by three independent reviewers. The reviewers appreciated the attention to an important topic. Based on the reviews, we are likely to accept this manuscript for publication, providing that you modify the manuscript according to the review recommendations.

Multiple reviewers pointed out the practical limitations of this approach, while highlighting the theoretical advancement made in this study. This limitation should be explicit in the revised version of your manuscript.

Sincerely,

Joel O. Wertheim

Associate Editor

PLOS Computational Biology

Daniel Beard

Deputy Editor

PLOS Computational Biology

[LINK]

Reviewer's Responses to Questions

**Comments to the Authors:**

Reviewer #1: The authors present an interesting and novel approach to phylogeny reconstruction. I would like to see a few changes prior to publication, some textual, but I'd also like to see two more kinds of simulation study to clarify the behavior under certain conditions.

Text changes:

1) Authors need to make it more prominently clear (maybe even in the abstract?) that this approach will estimate the tree and the haplotype frequencies, but not the actual sequence associated with each haplotype. This, along with the requirement that you must know, a priori, the number of haplotypes, is a rather severe limitation, and I struggle to think of a practical use for the proposed method. The typical case where we know the number of haplotypes in advance is when we explicitly mix a number of individually homogenous samples, but then we'd get a tree and frequencies, but have no way of saying which sample was associated to which tree tip, nor obtain a haplotype sequence for each sample.

My particular concern is that, as it is currently written, a biologist may read the manuscript and not appreciate this limitation, design a sequencing experiment without barcodes, run the software, and then become immensely frustrated that they can't recover the haplotype sequences with each of their mixed components.

2) "We implemented MCMCMC in AFPhyloMix, which reports a the tree topology and the tip frequencies which gives the highest value among all the resulting posterior probabilities along the computation."

This sounds like you're reporting a single sample from the chain? This is not very Bayesian in spirit, and could suffer from high variance, possibly differing substantially from one run to another (far more so than usual approaches to summarize MCMC chains). Given that tips are not tied to sequences in the usual way, so the "identity" of a tip can just drift, un-anchored, during sampling, I can see that you might have difficulties using other approaches to extract a consensus tree from the chain, but your chosen approach feels like it requires a little more justification and clarification.

3) "During the experiments, the following modified formula was found to have a better accuracy on the estimation of the tree topology and the tip frequencies:"

Could the authors explain this a little? How much better? Why did they try this formula in the first place? What is the explanation for its better performance?

4) "Whereas it is theoretically possible to simultaneously infer edge lengths and topologies, we have found that this does not deliver accurate results."

How inaccurate were they? This seems like strange behavior.

5) "The estimated result is regarded as correct if the tips on an estimated tree can be paired up with the tips on the actual tree satisfying the following conditions: (1) the difference between the predicted tip frequency and the corresponding actual tip frequency paired with is less than 0.01; and (2) their topologies are the same."

It isn't clear that this will produce a unique pairing of estimated to true tips, especially if the tip frequencies are similar (even at a single "cherry" - https://rdrr.io/cran/ape/man/cherry.html). If not, how do you efficiently choose the optimal tip-to-tip configuration to minimize later deviation estimates?

6) Typo: "AFPhyloMix adopts a Bayesian model of inference to estimate[s] the phylogeny"

7) Possible typo: "To further examine the [derivation] between the predicted tip frequencies 328 and the actual sample concentrations"

"deviation"?

8) Simulation request

This approach might not be especially robust to violations of the underlying model. The simplest of these is where rates vary over sites. Standard phylogeny inference is quite robust to such variation, but something that relies on SNP frequency distributions in the manner described here may not be. I suggest two kinds of studies to investigate such model violations:

A) Identical to your current simulation setup, but with varying levels of site to site variation in the Jukes Cantor mutation rate.

B) Take a set of "real" haplotypes that have evolved through a real evolutionary process (eg. a genome segment for a random set of mammals, or bacteria, or...), and use these as the basis of the simulation, where each simulation replicate involves the assignment of frequencies to each haplotype, and then the paired-end read simulation, etc. The "true" tree is the phylogeny inferred from the full haplotypes. I suggest finding a biological example where many genomes are available, so you can randomly select N each time, rather than using the same N over and over.

Note: perfect performance in these cases shouldn't be a requirement for publication - I just think it should be well-characterized exactly how things break down in these scenarios.

Reviewer #2: This manuscript addresses a critical problem in working with sequence data - separating haplotypes that are all sequenced together and are difficult to separate through pre-processing. This new approach addresses this challenge and also identifies the frequency of each haplotype and the relationship among haplotypes. The manuscript does a nice job explaining the method, and then using both simulations and an empirical example to test the method. However, there are some important aspects of this paper that would benefit from further explanation and rewriting.

First, I can see very important applications of this method that could be added to the introduction to greatly improve the motivation for the work. Primarily, understanding haplotype frequencies and the phylogeny of haplotypes in a mixed sample would seem to benefit naturally mixed samples, such as coinfections (e.g. malaria or SARS CoV-2). Focusing in the introduction dealing with quantities of data is irrelevant, while concern about pre-processing time and cost seems secondary to being able to sort our coinfections to which there is currently no solution. Rewriting isn't strictly necessary but would provide stronger motivation for the research.

The methods section begins with the expectations and calculations. However, this section then begins to explain some results (e.g. Figure 1C, 3B, ln 119+). The flow into the results is logical, except that the observed distribution is shown prior to any explanation of how it was generated. The simulations are not explained until the later section. I suggest significantly rearranging the Methods and Experiment sections so the theoretical expectations are provided first, then the simulations, then the results. Note that there are several other results integrated into methods (e.g. line 94, 212, 288) that should be moved as well.

I found the kangaroo data to be an odd example. It is clearly easy enough to tag these samples and get separate sequences for each sample. That said, it is an example that allows testing this method. To motivate its use better I suggest removing the discussion of kangaroo meat, which is completely irrelevant to AFPhyloMix, and just say this is a useful example where we could compare barcoded and not - you really could have picked any example where you mix samples at different concentrations.

Consider combining Fig 1B and C - overlaying the dots (expectation) on histogram (observation) would make the comparison clearer. Additionally, the points in 1B should be different shapes. B&C may need to be a separate figure in results. Similarly, Fig 3B is also results from simulations and may need to be moved, and overlaying A on B might make the comparison easier. The caption for Fig 1 should explain frequency v ratio so that that the caption can stand independently from the text.

The terms ratio and frequency are challenging, primarily because the authors make statements like "the expected SNP ratio ... would be... which is the frequency of tip A" (line 84). Ratio and frequency mean very specific things in this paper so the use of those words should be avoided otherwise. Ratio (an individual SNP) depends on sample proportion while frequency (across SNPs) depends on branch length.

Simulations were done with unrealistically high coverage. It would be helpful to know how well this approach works with lower coverage.

Why is the root sequence used as the reference for alignment? This seems unrealistic. This choice should be justified or a reference should be chosen from the "extant taxa".

The abstract and introduction refer to amplicon sequencing, while the simulations are for small genomes. I suggest changing the abstract and introduction to match what was actually done.

There are statements throughout such as Fig X shows... This phrasing is very awkward as it makes the results unclear. Please state actual results and refer the reader to the figure.

Fig 5B and C - consider normalizing these values. It's unclear what y-values are meaningful. Also, describe the quality of the results in the caption.

There are some minor typos and odd word choices throughout that I hope will be caught on rereading / in proofs.

Reviewer #3: Wong and colleagues present an interesting method to co-estimate the topology, branch lengths, tip frequencies and error rates (sequencing errors or others) from mapped reads, sequenced from a pooled DNA library, without barcoding for subsequent demultiplexing.

As a population geneticists working on ancient DNA, I am a bit outsider to the field, and need a few additional (mostly minor, I think) clarifications:

1) the applicability of the method. Authors say in the introduction that: "More importantly, there are

some samples where barcoding is impractical. For instances, rapidly evolving viruses

(e.g., Human Immunodeficiency Virus (HIV) and Hepatitis C Virus (HCV)) typically

exist as a collection of genetically diverse genomes within an infected individual.".

I truly understand your point here, but ...at the same time: "In our simulations, AFPhyloMix achieved at least 80% accuracy at recovering the phylogenies and frequencies of the constituent haplotypes, for mixtures with up to 15 haplotypes.". How many virus haplotypes do the authors expect to coexist during an infection? Probably way many more than 15, isn't it? How high is the mutation rate in viruses? Probably incompatible with an infinite sites model (?).

How to justify then the use of AFPhyloMix? What is the typical problem then were we can apply AFPhyloMix? Only mtDNA, given the poor reference assemblies of Y chromosomes? How expensive it is to sequence 10-15 mitochondrial genomes with barcodes? Not much really. Why do we need AFPhyloMix then? I am playing the devils advocate to motivate the authors to better justify the biological problems that could be addressed with AFPhyloMix in the introduction (I am sure there are). At the end of the day, this is what will condition the utility and the impact of their program.

2) the authors digest very well their equations making easy the reading and the understanding of the model, up until equation 6. They justify theoretically one equation, but suddenly "the following modified formula was found to have a better

accuracy on the estimation of the tree topology and the tip frequencies", without much further explanation, despite equation 6 being only an approximation. Could the authors please elaborate this a bit more, including a non-mathematical (more intuitive) description of equation 6 and why (they think) it works better (no simulations shown proving this). It might be obvious, but I am bit lost there, and I suspect that other readers will be as well.

3) what is the running time of the program? The authors discuss about the algorithm complexity, but a minimal description on the absolute scale would be informative. I guess AFPhyloMix is relatively fast given the number of simulations done, but readers might wonder if takes 1 week or more with eg. 15 long haplotypes.

4) Why not using the information of the pair ends (if paired-end data is available) to help defining the haplotype? Imagine there are linked mutations at both reads of each pair. This would provide additional information on the read<->haplotype assignation, which could especially help refining estimates of the haplotype frequencies at the tips.

5) What is the impact of the read length? In their empirical experiment, the authors test AFPhyloMix with reads that are 150bp-long and mtDNA , where the mutation rate is probably greater also in this species. I guess however that shorter reads and lower mutation rates reduce the chance of having more than one mutation per read, perhaps impacting the accuracy of the method. I am just asking whether the authors could clarify/elaborate a bit on this. Also, what is the substitution model or mutation rate used in their simulations?

6) Is there any chance to add an independent function, once topology, branch length and tip frequencies are estimated (ie. given the full phylogeny) to classify each read (or read pair) as belonging to one haplotype or another? That would be helpful for demultiplexing without barcodes. I understand this can be too much work for a revision, but I encourage the authors to think in implementing it after the publication (if the rest of concerns are satisfactorily addressed).

Thanks again for your efforts and congratulations for this nice approximation.

**Have the authors made all data and (if applicable) computational code underlying the findings in their manuscript fully available?**

Reviewer #1: Yes

Reviewer #2: Yes

Reviewer #3: Yes

PLOS authors have the option to publish the peer review history of their article (what does this mean?). If published, this will include your full peer review and any attached files.

Reviewer #1: No

Reviewer #2: No

Reviewer #3: No

Figure Files:

Data Requirements:

Reproducibility:

References:

---

## [Editor Report · Decision Letter 1]

1 Sep 2021

Dear Dr Wong,

We are pleased to inform you that your manuscript 'An assembly-free method of phylogeny reconstruction using short-read sequences from pooled samples without barcodes' has been provisionally accepted for publication in PLOS Computational Biology.

Best regards,

Joel O. Wertheim

Associate Editor

PLOS Computational Biology

Daniel Beard

Deputy Editor

PLOS Computational Biology

---

## [Editor Report · Acceptance letter]

9 Sep 2021

PCOMPBIOL-D-21-00630R1 

An assembly-free method of phylogeny reconstruction using short-read sequences from pooled samples without barcodes

Dear Dr Wong,

I am pleased to inform you that your manuscript has been formally accepted for publication in PLOS Computational Biology. Your manuscript is now with our production department and you will be notified of the publication date in due course.

With kind regards,

Andrea Szabo
